



# Effects of marine fuel sulfur restrictions on particle number concentrations and size distributions in ship plumes at the Baltic Sea

Sami Seppälä[1], Joel Kuula[1], Antti-Pekka Hyvärinen[1], Sanna Saarikoski[1], Topi Rönkkö [2], Jorma Keskinen[2], Jukka-Pekka Jalkanen[1], Hilkka Timonen[1]

[1]Atmospheric Composition Research, Finnish Meteorological Institute, Helsinki, 00560 Finland
[2]Aerosol Physics Laboratory, Tampere University, Tampere, 33100 Finland

*Correspondence to*: Sami D. Seppälä (sami.seppala@fmi.fi)

**Abstract**. Exhaust emissions from shipping are a major contributor to particle concentrations in coastal and marine areas. Previously, marine fuel sulfur content (FSC) was restricted globally to 4.5 m/m% but the limit was changed to 3.5 m/m% at

the beginning of 2012 and further down to 0.5 m/m% in January 2020. In sulfur emission control areas (SECA), the limits are stricter; FSC restriction was originally 1.50 m/m% but it decreased first to 1.00 m/m% in July 2010 and again to 0.10 m/m% in January 2015. In this work, the effects of the FSC restrictions on particle number concentrations (PNC) and size distributions (NSD) are studied at the Baltic Sea SECA. Measurements were made on a small island (Utö, Finland; 59° 46'50N, 21° 22'23E) between 2007 and 2016. Ship plumes were extracted from the particle number size distribution data, and the effects of the FSC

restrictions on the observed plumes as well as on the total ambient concentrations were investigated.

Altogether 42 322 analyzable plumes were identified during the 10-year measurement period. The results showed that both changes in the FSC restrictions reduced the PNC of the plumes. The latter restriction (to 0.10 m/m% in January 2015) decreased also the total ambient particle number concentrations, as a significant portion of particles in the area originated from ship plumes that were diluted beyond the plume detection limits. The overall change in the PNC of the plumes and ambient air was

27 and 32 %, respectively, for the total FSC change from 1.50 to 0.10 m/m%. The decrease in plume particle number concentration was caused mostly by a decrease in the concentration of particle sizes of ~ 35 - 134 nm. The latter restriction also reduced the count median diameter of the particles, which was probably caused by the fuel type change from residual oil to distillates during the latter restriction. The PNC was larger for the plumes measured in daytime compared to those measured in nighttime likely because of the photochemical aging of particles due to UV-light. The difference decreased with the reducing

FSC indicating that lower FSC has also an impact on the atmospheric processing of ship plumes.

## 1 Introduction

Particulate matter (PM) from shipping contributes to a significant fraction of PM[10], PM[2.5,] and PM[1] in European coastal areas (1-7, 1-14, and ≥11 %, respectively; (Viana et al., 2014). PM from shipping emissions can also be transported hundreds of kilometers inland (Lv et al., 2018), and have been linked to increased cardiovascular mortality and morbidity (Lin et al., 2018;

Partanen et al., 2013). The reduction of PM[2.5] emissions from shipping has been shown to reduce the negative health effects



(Barregard et al., 2019; C. Chen et al., 2019; Partanen et al., 2013; Sofiev et al., 2018). In multiple studies, shipping has also been reported to be a significant contributor to atmospheric particle number concentrations (PNC) both in the sea and coastal areas, especially in the ultrafine size range (Ausmeel et al., 2019; Gobbi et al., 2020; Karl et al., 2020; Kivekäs et al., 2014; Kukkonen et al., 2016; Zanatta et al., 2020). Atmospheric PNC has been considered to cause significant negative health effects

such as airway inflammation, impaired lung function, and cardiovascular mortality (Hennig et al., 2018; Strak et al., 2012). Sulfur dioxide ($SO_2$) is a gaseous air pollutant that is harmful to human health and the environment. Fossil fuels commonly contain sulfur that forms sulfur oxides ($SO_x$) in combustion (Arnold et al., 2006; Hoang and Pham, 2018). Most of the sulfur in diesel fuels are oxidized to $SO_2$ (Kozak and Merkisz, 2005; Walker, 2004), however, some $SO_2$ is further oxidized to $SO_3$ (3-5 %) (Cordtz et al., 2013). Already in the high-temperature exhaust, $SO_3$ can react with water molecules resulting in the

formation of gaseous sulfuric acid (GSA) (Arnold et al., 2012; Rönkkö et al., 2013). When the exhaust is cooled immediately after its emission to the atmosphere, the saturation ratio of GSA increases quickly leading to the homogenous nucleation of sulfuric acid and water (Arnold et al., 2006; Kozak and Merkisz, 2005) possibly supported by the exhaust of partly oxidized organic compounds (Arnold et al., 2012; Pirjola et al., 2015). After leaving the exhaust pipe, sulfuric acid contributes also to particle growth by condensing on existing particles formed in an engine, entrained aerosol particles, or newly formed

nanoparticles (Arnold et al., 2006; Riipinen et al., 2012). It should be noted that the oxidation of emitted $SO_2$ and the formation of sulfuric acid continues in the atmosphere leading also to the secondary formation of sulfuric acid and particulate matter (see e.g. Mylläri et al., 2016).

Fuel sulfur content (FSC) has a large effect on the formed particles and it has been reported to have a strong correlation with particle mass and number concentrations in the exhaust emissions of shipping (Alföldy et al., 2013; Diesch et al., 2013). The

effect is especially clear in the nanoparticle size range (Ushakov et al., 2013). Using fuels with lower FSC has been shown to reduce PM emissions (Lehtoranta et al., 2019) but FSC has also an influence on the chemical composition of exhaust particles. Typically, ship plume particles consist of organic matter (OM), black carbon (BC), and an inorganic fraction that is almost entirely composed of sulfuric acid (Kumar et al., 2013). At least sulfate and OM concentrations have been reported to be dependent on FSC ( Lack et al., 2009). Also, the emission factors (EF) of $SO_2$ and organic carbon (OC) have been observed

to decrease when fuels with lower FSC were used (Wu et al., 2019). In the study of Celik et al. (2019), OM was twice as large for fuels with FSC of 2.5 m/m% (mass per mass percentage) compared to FSC of 0.5 m/m%.

To reduce the harmful effects of ship emissions, restrictions for marine FSC have been set for Emission control areas (ECAs) and global marine transportation (IMO, 2008). Previously, the limit of FSC was globally 4.5 m/m% but it was changed to 3.5 m/m% at the beginning of 2012 and further down to 0.5 m/m% in January 2020. In sulfur emission control areas (SECAs)

specifically, the limit of FSC is lower and was changed from 1.50 m/m% to 1.00 m/m% in July 2010 and further to 0.10 m/m% in January 2015. In addition to these limits, the maximum FSC is limited to 0.10 m/m% (from the beginning of January 2010) for the berthed ships in EU ports (European Parliament and the Council of the European Union, 2005). The FSC limits and emissions control areas are described in more detail in (IMO Marpol Annex VI, Regulation 14). The decline of the real FSC of the marine vessels has been shown to follow the FSC restrictions well (Kattner et al., 2015; Pirjola et al., 2014), and therefore





the FSC limits can be expected to resemble the real maximum FSC of used fuels. However, Pirjola et al. (2014) reported that during the FSC restriction period of 1.50 m/m% (before July 2010), some vessels already used fuels with less than 1.00 m/m% FSC. This implies that the real FSC change has followed a gradual – rather than a step-wise – behavior at least for the July 2010 restriction enforcement.

The long-term effects of shipping emissions on atmospheric particle characteristics have been studied only in a few studies in

SECAs (Ausmeel et al. 2019; Kivekäs et al., 2014). In this study, a 10-year time series of particle number size distribution (NSD) data (from 2007 to 2016), measured near several busy shipping lanes on the island of Utö in the Baltic Sea SECA, was investigated. The effects of two changes in the FSC restrictions (from 1.50 to 1.00 and from 1.00 to 0.10 m/m%) on the PNC and NSD of ship exhaust plumes and ambient aerosol are discussed respectively. Additionally, the impacts of photochemical aging on ship exhaust plumes are investigated.

## 2 Experimental

### 2.1 Utö measurement site

The PNC and NSD data were measured at the atmospheric measurement station of the Finnish Meteorological Institute (FMI) at Utö between January 11[th], 2007, and December 31[st], 2016. Utö is a small island (0.81 km$^2$) at the southern edge of the Finnish archipelago in the Baltic Sea (59° 46'50N, 21° 22'23E) and it has a year-round population of 20 to 30 inhabitants. The

exact location of the measurement station on the island is shown in Fig. 1. The measurement station and location have been described in detail previously (Engler et al., (2007) and Laakso et al., 2018).

### 2.2 Ship routes and vessel types near Utö

Utö island is within the vicinity of busy shipping lanes. The closest shipping lane is on the western side of the Utö island, about 500 m away from the shore, and another busy shipping lane is 6 km south from Utö. Besides, Utö has a pilot station and harbor

with a regular passenger ferry connection to the mainland. The shipping routes near Utö are shown in Fig. 1. In Fig. 1, the data (white points) represents the Automatic Identification System (AIS) data from the Baltic Marine Environment Protection Commission (HELCOM). The AIS is a system that automatically produces and transmits information about vessels to other vessels and coastal authorities (IMO AIS transponders). This information includes identity, type, position, course, speed, and navigational status of the vessel and other safety-related information. Only a fraction of the AIS signals were plotted as the

total number of signals was excessive. In addition to estimating the presence and movement of vessels around Utö, the fractions of different types of vessels were also attained from the AIS data for this work. The AIS data were analyzed only for the vessels with International Maritime Organization (IMO) numbers as these are larger vessels, and smaller vessels without IMO numbers are not likely to contribute significantly to identified plumes, as shown by Kivekäs et al. (2014).

The sectors of the nearby shipping lane, distant shipping lanes, the harbor of Utö, and aging (aging discussed in Sect. 3.6) are

identified by colors in Fig. 1. While there is no clear upper limit to the distance from which the plumes may be arriving in the



measurement station, an limit radius (blue circle) was set at 25 km to count the number of AIS signals. This limit was chosen so that the harbors of other islands in northwest and northeast directions would not be included in the data as they would have increased the number of AIS signals from berthed ferries extensively.

The AIS markings from ships within the 25 km radius from Utö were divided into 10°-sectors (Fig. 2) and the activity of
different types of vessels was estimated in each sector. As HELCOM AIS signals are recorded every 6 minutes, the residence time (RT) of vessels occupying different sectors could be estimated by giving every AIS signal a 6 min RT. These RTs were plotted for the three FSC restriction periods and are presented in Fig. 2. Based on the HELCOM AIS data, the vessels were classified into six different categories best representing their specific type: cargo ships, large passenger ships, medium-sized passenger vessels, large work vessels, small vessels, and others.

The majority of the vessels around Utö were cargo and large passenger ships. While the amount of passenger vessel traffic and the distribution of traffic between the sectors remain almost the same between the different periods, the total amount of shipping, estimated from total RT of vessels, reduced from 71.1 (FSC < 1.50 m/m%) to 64.2 (FSC < 1.00 m/m%) and to 48.7 h d$^{-1}$ (FSC < 0.10 m/m%). The reduction was mostly caused by reduced cargo shipping activity. Notable is also that there were almost no vessels detected in the eastern directions (70-130°). The RT of vessels on the western side of the island was low
because the ships on the nearby shipping lane pass through the sectors quickly. The high number of passenger vessels at the direction of 330-340° was because of the signals from berthed ferries at the Utö ferry harbor.

**2.3 Particle measurement instrumentation**

The aerosol was sampled with a $PM_{2.5}$ inlet, which was positioned on the roof of the measurement container approximately 3 m above the ground. NSD and PNC were measured with a differential mobility particle sizer (DMPS). Before entering the
DMPS, the sample aerosol was dried with a Nafion dryer. The DMPS consisted of a 28 cm long Vienna-type differential mobility analyzer (DMA) and a continuous flow condensation particle counter (CPC). The theory and the response functions of DMA have been first described in Hoppel, (1978). The particle size range of the DMPS was 7-500 nm, and the size spectra were given in 30 size bins. The time resolution of the instrument was 5 minutes 20 seconds. During the measurements, the DMA and CPC were changed a few times due to malfunctions.

**2.4 Data processing method for plume identification**

First, the concentration peaks caused by the local sources (e.g. passing by of a tractor) were removed from the DMPS data. Identification of the peaks caused by the local sources was done by examining the time series of total number size distributions ($NSD_{tot}$) and total particle number concentrations ($PNC_{tot}$) visually as the concentration peaks from tractors, for example, were expected to last only tens of seconds whereas the durations of the ship plumes are measured in minutes. Secondly, the days
with only four or fewer measurement cycles were removed from further analysis to ensure the representativeness of the data. The remaining data was characterized as the cleaned data. The coverage of the cleaned data varied between ~26 - 96% depending on the year. The figure presenting the data coverage is presented in the supplemental data (Fig. S1).





Plumes emitted from the ships were separated from the cleaned DMPS data by using a modified version of the plume detection method developed and tested by Kivekäs et al. (2014). The method was originally developed for a scanning mobility particle

sizer (SMPS) data, but it is applicable also for the DMPS data. Since the method is described in detail in Kivekäs et al. (2014), only a short description of the method, the modifications made to it, and otherwise relevant information are given below.

The identification of plumes assumes that ship plumes entail significantly higher particle concentration than the natural background particle number concentration ($PNC_{bg}$). The $PNC_{bg}$ was defined as a sliding 25$^{th}$ percentile of 40 consecutive measurements and the excess particle number concentration ($PNC_e$) was calculated by subtracting the background

concentration from the $PNC_{tot}$. Maxima of the $PNC_e$ were examined if they fulfilled at least one of the two criteria set for the plume detection:

1)      The $PNC_e$ had to be at least 500 cm$^{-3}$ or

2)      the ratio of the $PNC_{tot}$ to $PNC_{bg}$ ($R_e$) had to be at least 1.5.

When the criteria was fulfilled, the $PNC_e$ was considered to be caused by a ship plume and is hereafter later in this study

referred to as particle number concentration in the plume ($PNC_{pl}$). In Fig. 3, a typical day with the plumes exceeding both limits is presented. Periods, when the red dotted line exceeds the black vertical line, are considered ship plumes. If two consecutive peaks exceed the detection limit and are separated by lower concentration, they are considered as two separate plumes. In terms of number size distributions, the studied NSDs are excess number size distributions ($NSD_e$), which are calculated by subtracting the background number size distributions ($NSD_{bg}$) from the total number size distributions ($NSD_{tot}$).

As shown by Kivekäs et al. (2014), fast and large changes in the $PNC_{bg}$ can cause an error in the detected plumes during these changes. These errors occurred as the sliding 25$^{th}$ percentiles used for defining $PNC_{bg}$ reacted roughly 10 measurement points too early to decreasing concentrations and 10 measurement points too late to increasing concentrations. This could have caused distortions to the plumes measured during the fast $PNC_{bg}$ change, or the fast changes could themselves have been counted as plumes. Therefore, the fast $PNC_{bg}$ changes were excluded from the data was divided into valid and invalid data periods by

counting absolute and the relative change rates of PNCbg and calculating the smoothed sliding average of six consecutive measurement points. When the absolute or relative change of the smoothed value was greater than $\pm$ 53 cm$^{-3}$ or $\pm$ 5 % respectively, the data was marked as invalid. These values were set so that they corresponded for 67 % of what was needed to define a plume. This validation procedure is described in detail in Kivekäs et al. (2014). Another factor that could have caused uncertainty to the detected plumes is the relatively long measurement cycle of the DMPS (5 min 20 s). However, the large data

set renders the effect of these uncertainties to be small. The errors related to the long measurement cycle are discussed in more detail in the supplemental material.



# 3 Results and discussions

## 3.1 Detected exhaust plumes

During the 10-year measurement period, a total of 71 811 ship exhaust plumes were observed, and, based on the data validation
procedure described in the experimental section, 42 322 (~ 59 %) of these were considered valid and representative. The
number and the frequency of the valid plumes are shown in Fig. 4 as a function of wind direction (in 10° sectors). The two
directions with the highest numbers of plumes (Fig. 4A) were 220-250° and 330-340°. The peak observed in the 330-340°
direction corresponds to the highest number of AIS markings and it is the direction where both the Utö ferry harbor and pilot
station are located. The second-largest number of plumes came from 220-250° direction, where only a few AIS signals were
observed within a 25 km radius. Similarly, a noticeable number of plumes were detected coming from a direction of 70-130°,
even though there were very few AIS signals in that direction. This indicates that some of the detected plumes had a distant
source, which for these directions, is likely to be the shipping lanes situated further away. However, all the emissions from
these distant shipping lanes are not expected to be detected as plumes as they probably diluted to be unidentifiable by the
detection method. Therefore, it is likely that the plumes detected from these directions have originally been the plumes with
high initial concentrations. When the numbers of plumes are normalized with the times of wind blowing from each direction
(Fig. 1C), the plume frequencies from different wind directions are more even (Fig. 4B). Exceptions are the direction of 50 -
110° with the longest distance to the shipping lane, and the direction of the Utö ferry harbor (330-340°), where the residence
time of the berthed vessels is high (Fig. 2).

## 3.2 Particle number concentrations of exhaust plumes

Particle number concentrations of the plumes ($PNC_{pl}$) were averaged annually, and the annual data were averaged over the
FSC restriction periods (Fig. 5). The results showed that the restrictions of FSC were effective; the average $PNC_{pl}$ decreased
from 1379 ± 30 (1.50 m/m%) to 1290 ± 170 (1.00 m/m%) and further down to 1010 ± 90 cm$^{-3}$ (0.10 m/m% ). Moreover, the
two lowest annual average $PNC_{pl}$ (940 ± 990 and 1070 ± 920 cm$^{-3}$) were observed during the 0.10 m/m% FSC restriction
period and three out of the four highest annual averages (1430 ± 2080, 1370 ± 1780, and 1355 ± 2120 cm$^{-3}$) were measured
during the 1.50 m/m% restriction period. Notable is also that the standard deviations are significantly lower for the annual
averages measured with the tighter FSC restriction periods indicating that especially the fraction of plumes with very high
concentrations have decreased. However, the highest annual average (1570 ± 2100 cm$^{-3}$), as well as the largest variability in
the $PNC_{pl}$ (the widest 25th and 75th percentile range), were observed in 2014, unexpectedly. Numerous intense nucleation
events occurred in 2014, however, they are likely to have been identified as invalid data and therefore do not affect the detected
plumes.

$PNC_{pl}$ attained in this work are similar to the concentrations of plume observed in similar studies; An median PNC of 750 and
1470 cm$^{-3}$ were measured during winter and summertime, respectively, in southern Sweden (Ausmeel et al. 2019). A noticeably
higher PNC concentration (2100 cm$^{-3}$) of ship plumes was measured with the airborne measurements right over the shipping





lane at the southern Baltic Sea by Zanatta et al. (2020) during the FSC restriction of 0.10 m/m%. Local meteorology, dilution,
and aging process of the plume, as well as the measured particle size range, may influence the variation of the results at least
to some degree.

In direct emission measurements, PNC of raw ship exhaust emission from idling have been reported being $432 \pm 1.83 * 10^8$
for idling and $24.6 \pm 0.26 * 10^7$ cm$^{-3}$ for HFO and MDO, respectively (Anderson, et al., 2015b) and in the range of $10^6$-$10^7$ cm$^{-3}$
after the dilution by a ratio of 8 (Karl et al., 2020). Therefore, fresh exhaust emission PNC can be expected to be around
$10^8$ cm$^{-3}$. Similar concentrations of $0.9$-$1.94 * 10^8$ cm$^{-3}$, depending on fuel and engine load, were found by Zhou et al. (2019).
As the concentrations measured from the plumes are in the range of $10^3$ cm$^{-3}$ in this study, the dilution ratio for the measured
plumes is estimated to be approximately $1:10^5$ to $1:10^7$.

### 3.3 Particle number size distribution of the exhaust plumes

Average number size distributions observed for the plumes (NSD$_{pl}$) are shown as a function of wind direction in Fig. 6. The
NSD$_{pl}$ were first divided into 10°-degree sectors and then averaged. These averaged NSD$_{pl}$ were interpolated to form a
continuous distribution. If the plumes had been plotted individually from every angle, the number of plumes from certain
angles would have been very low (even zero), and the individual plumes would have had a disproportional effect on the
distribution.

The plumes with the highest PNC$_{pl}$ arrived from the sector of 190-240° and the plumes with the lowest concentrations from the
sector of 50-90° (Fig. 6). Near Utö, the direction of 190-240° was the direction with the highest shipping activity excluding
the berthed ferries at Utö harbor (Fig. 2). Especially the cargo vessel activity in that direction was high. In contrast, in the
direction of 50-90° the shipping activity was almost nonexistent near Utö. PNC$_{pl}$ from the direction of the ferry harbor of
Utö (310-340°) was generally lower than PNC$_{pl}$ from the plumes arriving from 190-240°, despite many of the plumes that were
expected to originate from the berthed ferries at the harbor (Fig. 1D). This might be due to the small size of the berthed ferries,
which produced smaller amounts of particulate matter. The small ferries also operated on distillate fuels during all the FSC
restrictions. Distillate fuels have been shown to produce lower particulate emissions on lower engine loads (Zhou et al., 2019).
Stricter FSC limits were seen as a decreased PNC$_{pl}$ in the direction of nearby ship traffic (Fig. 6). The effect was visible after
both restriction changes, but the decrease was significantly larger after the change of FSC restriction from 1.00 to
0.10 m/m%. Besides PNC$_{pl}$, the change of FSC limit from 1.00 to 0.10 m/m% was observed to reduce the particle size. To the
knowledge of the authors, no previous studies are reporting the effects of FSC on particle size and concentrations from
atmospheric exhaust plume measurements. Furthermore, the results from direct emission measurements are divided in
literature; Zetterdahl et al. (2016) reported that the stricter FSC limits reduce the particle size, but not the produced PNC$_{pl}$,
whereas, for example, Streibel et al. (2017) found lower emission factors for PNC$_{pl}$ and larger particle size when using distillate
fuels instead of the HFO.
In terms of plume number size distributions, it is notable that most of the plume particles were smaller than 100 nm in diameter
and case of the FSC restriction of 0.10 m/m% even smaller than 50 nm (Fig. 6). Therefore, the contribution of ship emissions




to particle mass is likely to be small, and the characterization of ship plumes should be based on number concentrations rather than mass concentrations in plumes. This has also been proposed by Viana et al. (2014).

Count median diameters (CMD) of particles in plumes were calculated and are shown as a function of wind direction in Fig. 7.
The CMD remained almost unchanged during the change from the FSC restrictions of 1.50 to 1.00 m/m%. However, the FSC restriction change from 1.00 to 0.10 m/m% significantly reduced the CMD of particles arriving from all wind directions. Overall, the largest CMD of the plumes was observed from eastern and southeastern directions (60-170°), where the distance to the shipping lanes is large, and the smallest CMD for the plumes from north to northwester directions (300-330°), which is the direction of Utö ferry harbor. This variation of the CMD from a minimum of 36, 39, and 31 nm to the maximum of 58, 55,
and 51 nm for the FSC restrictions of 1.50, 1.00, and 0.10 m/m%, respectively, implies that particles either grow significantly or small particles evaporate when they are transported in the atmosphere. Zanatta et al. (2020) found similar behavior of increasing particle size as the age of the plumes increased. They proposed this to be a result of coagulation and dilution-controlled processes. The atmospheric processing of plume particles is discussed more in Sect. 3.6.

In direct emission measurements, the FSC has been related especially to produced nanoparticles, and the effect on the larger
particles has been observed to be smaller (Ushakov et al., 2013). Therefore, in this work, particles were divided into two size ranges: 7-134 nm and 155-499 nm, and their concentration changes were studied separately. The two size ranges were calculated from the DMPS bins of 1-21 (midpoints of the size bins were 7-134 nm ) and 22-30 (midpoints of the size bins were 155-499 nm).

Based on the two size ranges, FSC restrictions were more effective in 7-134 nm particles. The $PNC_{pl}$ of the smaller particles
decreased clearly, while that of the larger particles remained almost unchanged when the FSC limit became tighter (Fig. 8). $PNC_{pl}$ were also calculated using slightly different size ranges and figure with ranges, 7-116 nm, and 134-499 nm is presented in the supplemental material (Fig. S2). The figure shows that if the lower limit for the larger particle size is smaller than 155 nm, the change in the FSC restriction is also seen in the larger size class as reduced $PNC_{pl}$ after the FSC change from 1.00 to 0.10 m/m%. Therefore, the effect of sulfur restrictions seems to be mostly limited to the $PNC_{pl}$ in the particle sizes smaller
than 134 nm.

The $PNC_{pl}$ in the smaller size range (7-134 nm) was observed to be highest when the wind was blowing from the direction of 190-340° (Fig. 8), where there is plenty of shipping activity nearby e.g. nearby shipping lane and Utö harbor (see Fig. 1D). In contrast, the $PNC_{pl}$ of the larger size range (155-499 nm) was highest when the wind was blowing from the direction of 60-210°, where there is a significant amount of ship traffic at longer distances from the measurement site (see Fig. 1A). This
supports the assumption that particles grow in the atmosphere to larger particle sizes through atmospheric processing (e.g. coagulation or condensation) and the increased CMD is not only due to the evaporation of small particles. For the larger particle size range (155-499 nm), the similarity of the shapes of the $PNC_{pl}$ patterns in different FSC restriction periods indicated also that the ship fleet was responsible for the plumes and the shipping routes have stayed relatively constant between different restriction periods. This strengthens the assumption that the differences observed in this study were caused by the FSC
restriction changes and not the change in the ship fleet.



### 3.4 Plume number size distributions for shipping lane sectors

To investigate the number size distributions of plumes ($NSD_{pl}$) in more detail, two sectors, distant shipping lanes (70-130°) and nearby shipping lanes (280-300°), were chosen for a separate analysis. The sectors were selected so that the distant shipping lane represented plumes that were transported longer distances whereas nearby shipping lanes represented fresher plumes. These assumptions of plume ages are based on the locations of the shipping lanes (Fig. 1), the CMDs of the plumes (Fig. 7), and the dependency of particle concentrations on wind directions (Fig. 8). Even though the CMDs for the plumes from the harbor of Utö (320-350°) were the smallest, they were not included as a large number of these plumes were expected to be originated from the berthed ferries operating on fuel with FSC < 0.10 m/m% during all three FSC restriction periods. In the case of distant shipping lanes (70-130°), it was ensured from Fig. 2 that there were no additional vessels between the measurement station and the shipping lanes (a distance of ~25 km). In the nearby shipping lanes sector, almost all of the AIS markings were from cargo vessels from the nearby shipping lane (a distance of ~1-2 km).

For both the nearby and distant shipping lane sectors, the shapes of the $NSD_{pl}$ remain nearly unchanged after the change from the FSC restriction of 1.50 to 1.00 m/m% (Fig. 9). In the distant shipping lanes sector, the CMD shifted from 51 to 49 nm, and in the nearby shipping lanes sector from 40 to 41 nm, when the FSC restriction changed from 1.50 to 1.00 m/m%. After the FSC restriction changed from 1.00 to 0.10 m/m%, the shape of the $NSD_{pl}$ altered more clearly as the CMD of the particles shifts to significantly smaller particle sizes; from 49 to 44 nm in the distant shipping lanes sector (70-130°) and from 41 to 32 nm in the nearby shipping lane sector (280-300°). The reduction of the particle size with decreasing FSC has also been reported by Zetterdahl et al. (2016), who observed the decrease in the particle size mode from 34 to 19 nm in the raw exhaust gas for the ships using fuels with FSC of 0.48 and 0.092 m/m%, respectively.

The $NSD_{pl}$ from the distant shipping lanes sector (70-130°) were observed to be bimodal consisting of a nucleation mode with larger particle number concentrations at ~30-60 nm and a soot mode with smaller number concentrations at ~100-200 nm. The mode in the particle sizes larger than 100 nm in ship emission exhaust has been identified to be a soot mode (Ntziachristos et al., 2016). The similar shape of the NSD has been attained for ship diesel exhaust emissions from direct emission measurements by Corbin et al. (2020) having similar proportions between the nucleation and soot modes and with the nucleation mode particle size being slightly smaller ~35 nm. This might be because the plumes measured in this study had time to grow in the atmosphere by coagulation, as indicated in Fig. 7. Increasing particle size via coagulation has been also reported by (Petzold et al., 2008). In the literature, NSDs of ship plumes have been typically observed to be unimodal (Celik et al., 2019) or bimodal (Petzold et al., 2008), whereas emissions measured directly from the engine exhaust are often bimodal (Anderson, Salo, & Fridell, 2015a; Ushakov et al., 2013). In addition to the former, also trimodal NSDs measured directly from the marine engine exhaust have been reported by (Tian et al., 2014), who found a distinct volatile mode at 15 nm and two combustion-related modes at 38 and 155 nm.



### 3.5 Total particle concentrations and size distributions in ambient air

The main focus of this work was to investigate the effects of the FSC restrictions on ship plumes, but in addition to this, shipping emissions are expected to contribute also to the ambient total particle number concentrations ($PNC_{tot}$, including

background concentrations, $PNC_{pl}$, and the part of $PNC_e$ that was not considered as valid plumes) at Utö. The annual average $PNC_{tot}$ was calculated for each year (2007-2016) and averaged for the different FSC restriction periods (Figure 10). The average $PNC_{tot}$ was almost the same during the FSC restriction of 1.50 (2910 ± 110) and 1.00 m/m% (2960 ± 1140 cm$^{-3}$) However, after the FSC restriction was set to 0.10 m/m%, the average $PNC_{tot}$ decreased significantly, to 1970 ± 200 cm$^{-3}$, probably because of the switch between heavy residual fuels and lighter distillates. The distillate fuels with lower FSC have

been shown to produce lower particle emission (Zhou et al., 2019).

Despite the more stringent FSC restriction, the annual average $PNC_{tot}$ was as high as 4840 ± 5270 cm$^{-3}$ in 2014. This was attributed, at least partially, to the numerous intense nucleation events occurring in that year (see example of nucleation event in supplemental material Fig. S3). These nucleation events increased the concentrations of small particles by several times, and the concentration levels stayed elevated for extended periods. These events fulfilled the criteria for the nucleation event

set by Dal Maso et al. (2005): appearing of a new mode to the size distribution in the nucleation mode size range, this mode has to prevail over several hours and show signs of growth. These criteria differentiate from the ship plumes as the ship plumes last much shorter periods and do not show significant growth. Moreover, the plumes had even the peak of $NSD_{pl}$ at larger particle sizes (~30-60 nm) compared to the definition of nucleation mode (3-25 nm) by Dal Maso et al., (2005). The existence of the nucleation events is also indicated by the much larger variability in the $PNC_{tot}$ values in 2014 than for the other years

(Fig. 10).

To further investigate the effects of FSC restriction changes on the ambient aerosol at Utö, the average total number size distributions ($NSD_{tot}$) were plotted as a function of wind direction for the three FSC restriction periods (Fig. 11). The highest $PNC_{tot}$ was observed during the FSC restrictions of 1.50 and 1.00 m/m% with the wind directions between 80-120°, 160-230°, and 260-290°. During the FSC restriction of 0.10 m/m%, the $PNC_{tot}$ was elevated also in the same directions, however, the

average $PNC_{tot}$s were significantly lower. The highest $PNC_{tot}$ was observed in the size range of 40 to 100 nm, which is a typical size range for ship emission particles (Alanen et al., 2020; Celik et al., 2019; Jonsson et al., 2011; Petzold et al., 2008; Westerlund et al., 2015). All these sectors with elevated $PNC_{tot}$ correlated well with the distant shipping lanes (directions of distant shipping lanes estimated from figure 1A; 80-110°, 170-250°, 270-290°), indicating that a large fraction of the measured background particle concentrations in the size range of 7-500 nm originated from the distant, diluted ship plumes. A large

fraction of $PNC_{tot}$ being from diluted ship plumes is also supported by the fact that the elevations of $PNC_{tot}$ in directions of the distant shipping lanes decrease with the more stringent FSC restriction. However, the effect of FSC restriction changes on the $PNC_{tot}$ might be slightly exaggerated, as also the shipping activity in the immediate area of Utö has decreased (Fig. 2). The change of ship traffic on the distant shipping lanes, however, was not directly studied and might differ from the change seen





in Fig. 2. In a previous study of Kecorius et al. (2016), it was found that ambient $PNC_{tot}$ increased 26-53 % when the air mass was transported over the Baltic Sea, and most of the increase was related to shipping traffic.

The ship exhaust plumes arriving from long distances are not observed as individual plumes as they do not cause detectable peaks in the data due to excessive dilution. Excessive dilution causing ship plumes to become undetectable is presented also in other studies: ship plumes were undetectable when plumes arrived from at least 100 km (Kecorius et al., 2016). Petzold et al. (2008) estimated that in 24 hours, the ship exhaust plumes get entirely mixed with the marine boundary layer. Chen et al.
(2005) found that due to the dispersion, the lifetime of a plume was $2.5 \pm 0.6$ hours when the detection criteria were the same as the second criteria used in this study ($R_e > 1.5$). As the FSC restrictions seem to affect both $PNC_{pl}$ and $PNC_{tot}$, they have a larger effect on the air quality than only the direct effects measured from nearby ship exhaust plumes would assume.

**3.6 Photochemical aging of the plumes**

Photochemical aging changes the physical and chemical properties of the ship plume particles. Sulfur content in fuel affects
$SO_2$ emitted from marine engines, but Zetterdahl et al. (2016) measured also reduced volatile organic compound (VOC) emissions related to the smaller FSC. However, in contrast to decreased secondary organic aerosol (SOA) precursor observations ($SO_2$ and VOC), Wu et al. (2019) observed an increase in SOA formation due to changing from residual fuel oils to distillate fuels. At Utö, the $SO_2$ rich ship emissions from the Utö harbor area have been previously linked to being a possible cause of local nucleation events (from the secondary aerosol formation) especially in the size range of 10-30 nm (Hyvärinen
et al., 2008).

In this study, the effect of photochemical aging on the exhaust plumes was examined by comparing the number size distributions ($NSD_{pl}$) and volume size distributions ($VSD_{pl}$) measured in nighttime (total radiation intensity $I_{tot} < 0$ W m$^{-2}$) and in the daytime when $I_{tot} > 200$ W m$^{-2}$. The $NSD_{pl}$ and $VSD_{pl}$ were calculated as averages in 10°-sectors and the number of plumes exceeding the limit values per angle are presented in the supplemental material (Fig. S4). Only the sectors of 220-260°
and 320-350° had enough data for both nighttime and daytime plume analysis by exceeding 50 plumes per 10°. With fewer plumes per sector, noise in the data was high (Fig. S5). Moreover, the sector of 320-350° was the direction of the ferry harbor of Utö and was therefore excluded from further analysis.

Night- and daytime $NSD_{pl}$ and $VSD_{pl}$ differed from each other during all the three FSC restriction periods (Fig, 12). A large fraction of the change between day and night might be related to the different meteorological conditions and ship traffic during
day and night. Assuming that the wind speed, humidity, temperature, and other meteorological parameters, as well as the composition of the ship fleet, stayed relatively constant between the restriction periods, the remaining variable is FSC, and the differences between the restriction periods are caused by the changes in the FSC limits. These assumptions cause large uncertainty in this result and the results are therefore only considered qualitatively.

The most striking difference between the day- and nighttime $NSD_{pl}$ was that the size distribution maximum shifted to the larger
particle size in the daytime (Fig. 12). The shift was largest for the FSC limit of 0.10 m/m% and smallest for the FSC limit of 1.50 m/m%. Moreover, during the FSC limit of 0.10 m/m%, the maximum of the $NSD_{pl}$ was clearly at the smaller particle size





than the maxima at 1.00 and 1.50 m/m% at nighttime whereas in the daytime the maximum was almost independent of the FSC limit.

$VSD_{pl}$ was bimodal during all three restriction periods. The mode at the particle size of ~90 nm was smaller with relation to

the mode at ~300 nm, the difference between the modes being especially noticeable during the FSC restriction of 0.10 m/m%. Similar to the $NSD_{pl}$, the mode at ~90 nm shifted to the larger particle size in daytime compared to that in the nighttime. Again, the shift was largest for the FSC of 0.10 m/m% and smallest for the FSC of 1.50 m/m%.

When the effects caused by dilution to the $PNC_{pl}$ are excluded, the remaining factors in the atmosphere influencing $PNC_{pl}$ are coagulation and new particle formation (Celik et al., 2019). Also, condensation and evaporation can change $NSD_{pl,}$ and in the

case of evaporation, it can affect the $PNC_{pl}$ by evaporating particles completely or reducing particle size below the detection limit of the measurement instrument. Celik et al. (2019) suspected that during nighttime, coagulation reduced the particle number emission factors, but they did not find any correlation between plume age and plume particle number emission factor ($E_n$) during the daytime. It was proposed that this could be because new particle formation during daytime overrides the effects of coagulation. Shiraiwa et al. (2013) have presented that in photochemical aging, semi-volatile organic compounds (SVOC)

equilibrate with the particle population and the majority of SVOC condense on the largest particles because of their greater absorption capacity and reduced equilibrium vapor pressure on the surface of the particle (the Kelvin effect). They found that the maxima of the NSD got narrower and shifted to a larger particle size at the diameter of 200 nm in 20 h of photochemical aging and depleted basically all particles below 100 nm. This kind of behavior is an extreme case but could be another plausible explanation for the observed behavior of $NSD_{pl}$ seen in this work.

Table 1 represents total $PNC_{pl}$ and volume concentration of the plume particles ($VC_{pl}$) for the three different FSC restrictions for the day- and nighttimes. Overall, the $PNC_{pl}$ was higher during nighttime plumes than during the daytime. The difference was largest for the FSC restriction of 1.50 m/m% and the smallest for the 0.10 m/m% restriction. Each change in the FSC restrictions was observed to reduce the $PNC_{pl}$ both in the day- and night time, however, the dependency of the $PNC_{pl}$ on the FSC was greater in the nighttime, and therefore the night-day difference in the $PNC_{pl}$ decreased with decreasing FSC. In

contrast to $PNC_{pl}$, the $VC_{pl}$ was higher during the daytime plumes but similar to the $PNC_{pl}$, the $VC_{pl}$ reduced with the stricter FSC limit. For the $VC_{pl,}$ the difference between the day- and nighttime concentrations were less dependent on the FSC limit than for the $PNC_{pl}$.

Higher total $VC_{pl}$ during daytime than during nighttime indicates the condensation of gaseous matter on particles. Meanwhile, the $PNC_{pl}$ was lower in the daytime, which might indicate either coagulation or evaporation of small particles. The decrease of

$PNC_{pl}$ and increase of $VC_{pl}$ in daytime diminishes as the FSC reduces implying that the higher FSC increases either the coagulation or evaporation of small particles and the condensation of larger particles. Coagulation cannot explain the higher number of smaller particles observed in Fig. 12, but the evaporation of liquid matter from the surface of medium-sized particles exposing the small solid core particles could explain this. The evaporated material from these particles, and the wholly evaporated particles together with other VOCs in plume condensate over larger particles.



## 4 Conclusions


This study investigated the effects of the fuel sulfur content (FSC) restrictions on the number size distributions ($NSD_{pl}$) and particle number concentrations ($PNC_{pl}$) at the Utö island in the Baltic Sea during the 10-year time-period (2007-2016). The focus was both on the changes in the ship exhaust plumes and the ambient aerosol. In general, FSC restrictions were observed to have a clear impact on particle number concentrations and size distributions in the marine environment next to the shipping

lanes. Besides, the FSC limitations affected the size distributions and the growth of particles during the plume aging.

The overall change in FSC from 1.50 to 0.10 m/m% decreased the $PNC_{pl}$ and total particle number concentration ($PNC_{tot}$) by 27 and 32 %, respectively. In both cases, the largest decrease was observed after the latter FSC restriction change from 1.00 to 0.10 m/m%, likely because the fuel type was changed from residual oil to distillates.

Particle size distribution in the plumes also changed after the FSC restriction change from 1.00 to 0.10 m/m%. The FSC

restriction affected especially the concentrations of the small particles by reducing the particle number concentration in particle sizes ~ 35-134 nm, whereas particles with $D_p$ > 155 nm remained mostly unaffected. Due to the small size of particles in ship exhaust plume emissions (CDM typically 30-60 nm), the contribution of ship emissions to $PM_{2.5}$ is likely to be small and the effect of ship emissions should be characterized based on PNC rather than PM.

In addition to the isolated ship plumes, there was a decrease in the total ambient particle number concentrations ($PNC_{tot}$) with

the stricter FSC limits. When the wind was blowing over distant shipping lanes, the $PNC_{tot}$ was significantly elevated because of large numbers of diluted ship plumes arriving in Utö. This observation indicates that shipping is a major source of particles in the marine environment, even though the distance to the shipping lanes can be substantial. Jonson et al. (2019) proposed that the contribution of shipping related particle emissions to $PM_{2.5}$ would be negligible after the FSC restriction of 0.10 m/m%. However, based on the results of this study, this does not seem to be the case, as the reduction of $PNC_{pl}$ and particle size were

moderate and the concentrations in the largest particle sizes remained unchanged. However, the largest particle sizes can be affected by other sources than shipping in Utö, for example, long-range transported particles from other combustion sources in the mainland.

Count median particle size (CMD) was found to increase with the increasing residence time of ship exhaust plumes in the atmosphere. CMD was larger for the particles in plumes that traveled long distances. Furthermore, particle volume

concentration was larger for the plumes measured in daytime compared to the plumes measured in nighttime likely due to the photochemical aging of particles due to UV-light. This difference between day and nighttime decreased with the reducing FSC. This indicates that lower FSC might reduce the condensation of gaseous matter on larger particles and either diminishes coagulation or evaporation of smaller particles. This result includes several uncertainties and therefore needs further investigation.

We note that during a decade long measurement period, also the ship traffic somewhat decreased in the vicinity of UTÖ and it is possible that the decrease in activity, in addition to FSC limits, has affected the observed ambient particle number concentrations. However, especially the change from 1.00 to 0.10 m/m% FSC was observed to cause a clear change also in



ambient particle concentrations. Also, the decrease in ship traffic is assumed to be gradual and is not expected to cause this kind of step-change. Also, the amount of ship traffic does not affect the measured concentrations and number size distributions

of plumes as these average values do not dependent on the number of measured plumes.

As conclusions, the FSC restrictions can be expected to cause impacts on the environment and human health due to the lower particle number concentrations in the ultrafine particle size range. Concentration reduction of ultrafine particles is especially effective in reducing negative health effects as the respirable fraction of ultrafine particles is high (Heyder, 2004) and the lower fraction of the respiratory tract is more sensitive to pathogen numbers compared to upper parts of the respiratory tract (Thomas,

2013). Reduction of particle concentrations in this study in the size range of ~ 35-134 nm has significant contributions to cloud formation as 40-120 nm (electrical mobility diameter) has been identified as the most crucial size for cloud condensation nuclei as larger particles are almost always activated to form cloud droplets and the particles smaller than 40 nm are not activated with realistic supersaturations (Dusek et al., 2006). Therefore, ships produce fewer cloud condensation nuclei, and this might have a large warming effect on the climate as marine clouds usually have a cooling effect on the atmosphere. Increasing

average cloud droplet concentrations from 100 to 375 $cm^{-3}$ has been presented to have more than double of a cooling effect on the climate than the warming caused by the doubling of $CO_2$ (Latham et al., 2008). Therefore, even a slight decrease of cloud condensation nuclei over sea areas might have a strong cooling effect on the climate.

*Data availability:* Data are available upon request from the corresponding author Sami Seppälä (sami.seppala@fmi.fi).


*Author contributions.* Sami Seppälä handled the formal analysis, software, visualization, and writing- original-draft, Joel Kuula, Antti-Pekka Hyvärinen, Sanna Saarikoski, Topi Rönkkö, Jorma Keskinen, Jukka-Pekka Jalkanen contributed to ideas, review and editing of the article, Hilkka Timonen acted as a supervisor and contributed to reviewing and editing of the article.

*Competing interests.* The authors declare that they have no conflict of interest


*Acknowledgements.* This work was financed by the European Research Infrastructure for the observation of Aerosol, Clouds, and Trace Gases (ACTRIS) and from the European Union's Horizon 2020 Programme Research and Innovation action under grant agreement No 814893, SCIPPER.

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





**Figure 1: A) The location of the Utö island and location of the vessels (every 10 000th shown) with IMO numbers in 360 000 km² area around Utö. B) Location of the vessels (every 500th shown) with IMO numbers in 10 000 km² area around Utö. C) Percentages of the time that wind is blowing from different directions during valid measurement periods divided into 10°-sectors. D) Location of the vessels (every 10th shown) with IMO numbers in 100 km² area around Utö. White dots in pictures A, B, and D, mark the locations of vessels. Distant shipping lanes sector, aging sector, nearby shipping lane sector, and harbor of Utö are marked with colored lines. Base map and data from OpenStreetMap and OpenStreetMap Foundation (OSMF). Data from © OpenStreetMap contributors is licensed under the Open Data Commons Open Database License (ODbL) by the OSMF.**







**Figure 2: Ship types around Utö during the different FSC restriction periods A) FSC < 1.50 m/m%, (11.1.2007-30.6.2010), B) FSC** 
**< 1.00 m/m%, (1.7.2010-31.12.2014), C) FSC < 0.10 m/m%, (1.1.2015-31.12.2016). Ship types were classified as cargo ships, large passenger ships, medium-sized passenger vessels, large work vessels, small vessels, and others. On the y-axis, RT is the average residence time in hours per day of ships in the category on each of the 10°-sectors given on the x-axis. Distant shipping lanes sector, aging sector, nearby shipping lane sector, and harbor of Utö are marked with vertical line pairs.**





**Figure 3: An example of the time series of A) number size distribution of excess particles (NSD$_e$), B) particle number concentration of excess particles (PNC$_e$) and C) the ratio of total particle number concentration to the background particle number concentration (R$_e$) for a single day. In B) – C) the plume detection limits have been marked with black lines.**



**Figure 4: A) Number of valid plumes and B) the frequencies of the plumes per hour from different wind directions. Distant shipping**
**lanes sector, aging sector, nearby shipping lane sector, and harbor of Utö are marked with vertical line pairs.**


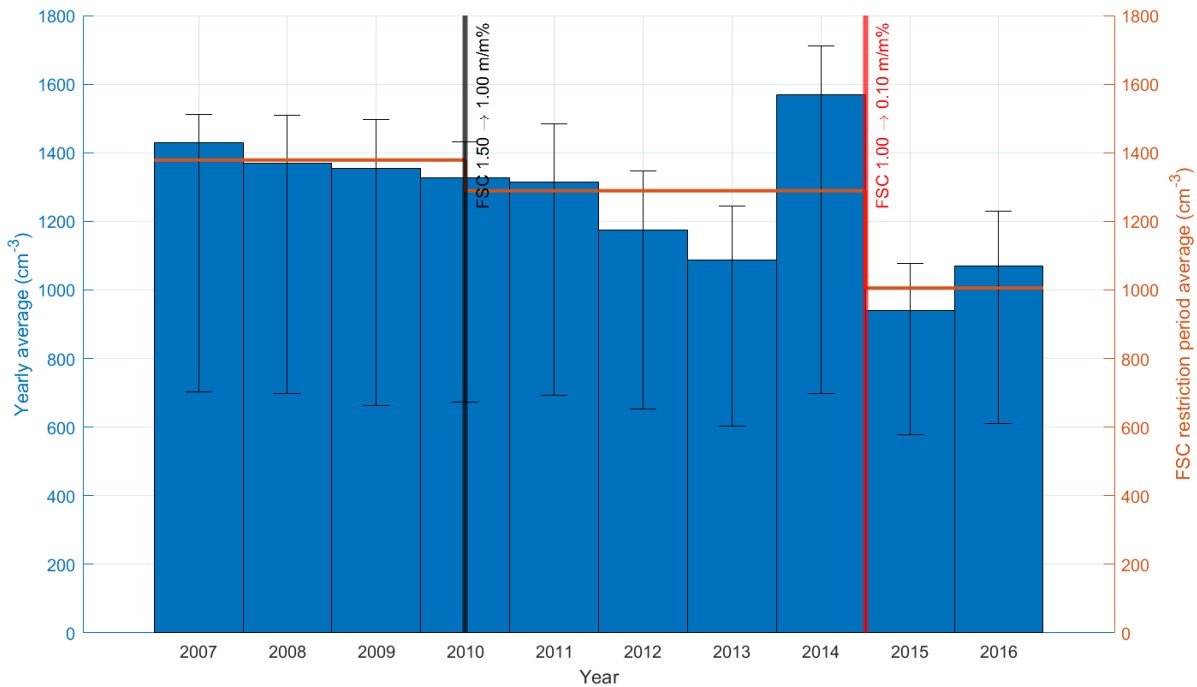

**Figure 5: Annual average particle number concentrations of plumes (PNC$_{pl}$, blue bars) and the average PNC$_{pl}$ during different FSC restriction periods (orange lines). Changes in the FSC restrictions are marked with the black and red vertical lines. The 25$^{th}$ and 75$^{th}$ percentiles are marked with the black error bars.**





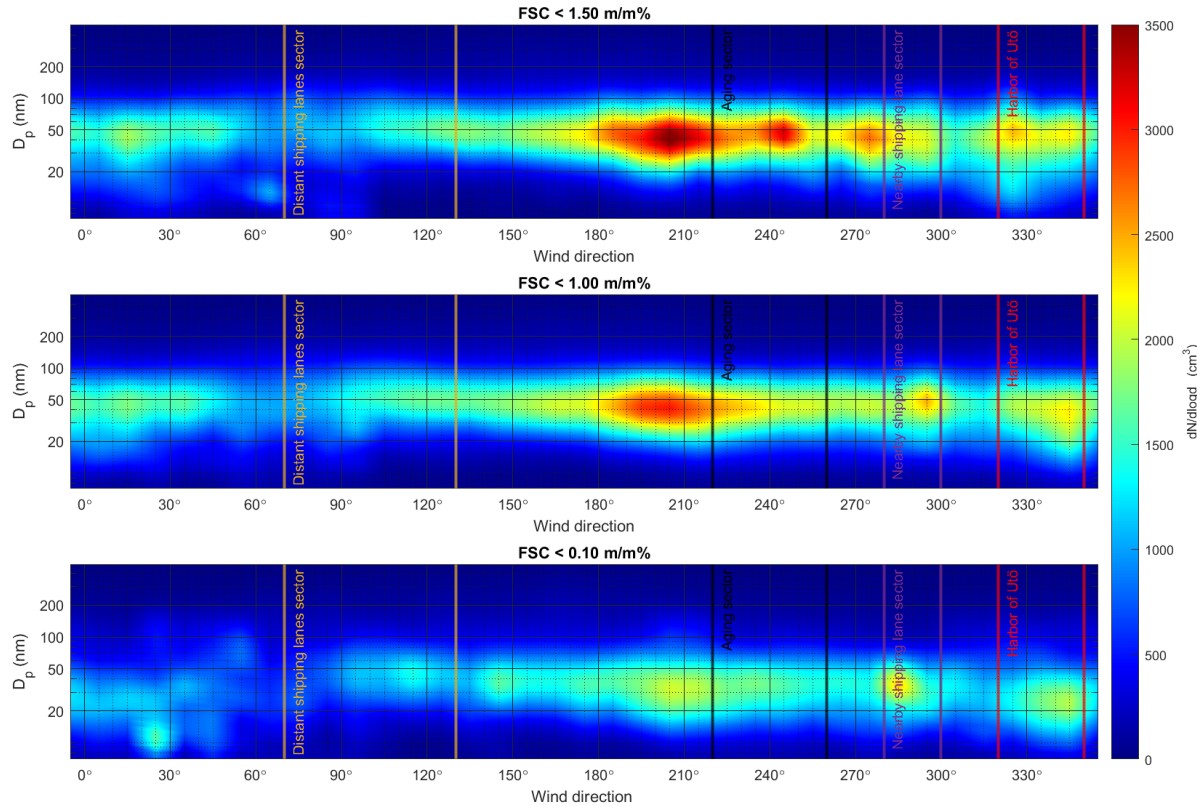


**Figure 6: sectors) for different FSC restrictions. Wind directions are shown on the x-axis, particle mobility diameters ($D_p$) on the y-axis, and normalized particle number concentration ($dN/dlogd_p$) with colors. Distant shipping lanes sector, aging sector, nearby shipping lane sector, and harbor of Utö are marked with vertical line pairs.**





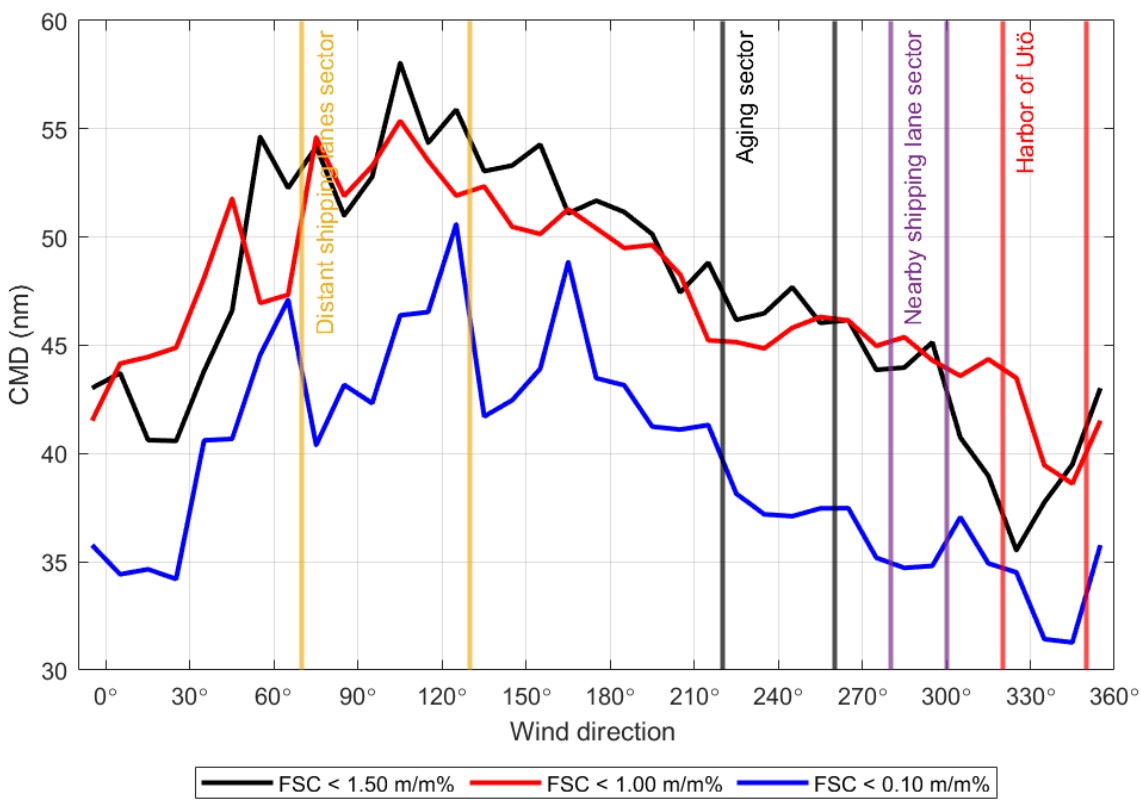

**Figure 7: Average count median diameters (CMD) of particles in plumes originating from different directions during the different fuel sulfur content (FSC) restrictions. Distant shipping lanes sector, aging sector, nearby shipping lane sector, and harbor of Utö are marked with vertical line pairs.**





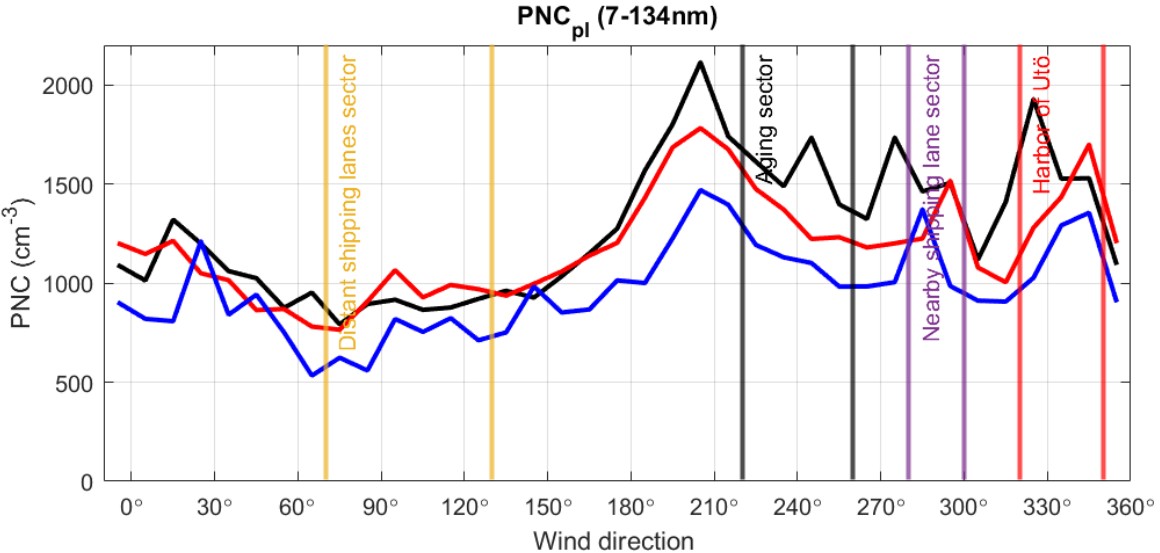

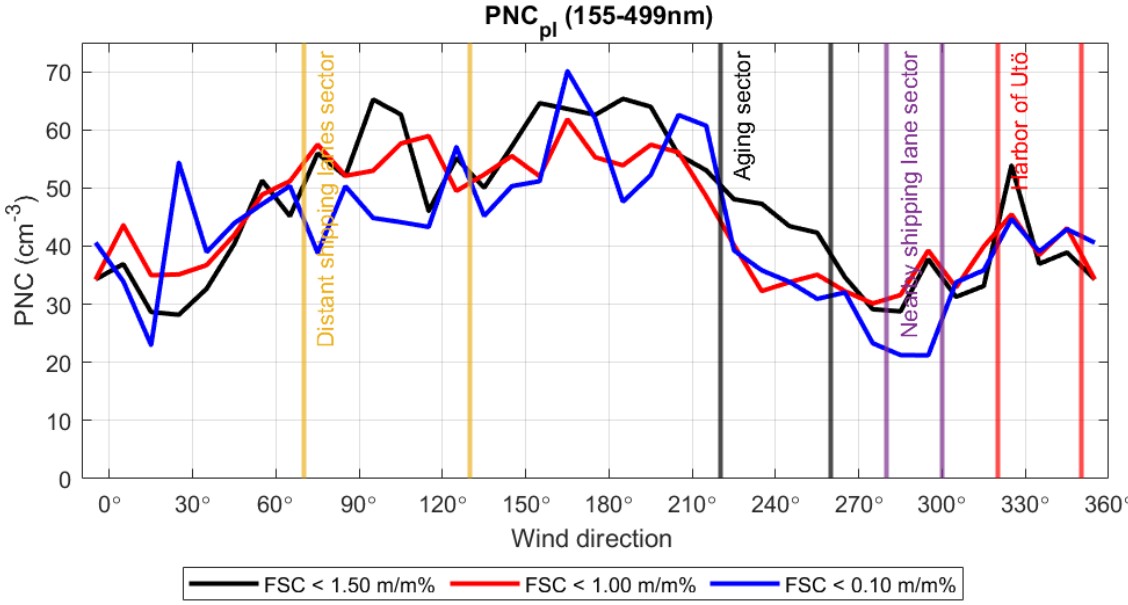

**Figure 8: Particle number concentrations of the plume particles (PNC$_{pl}$) in size range 7-134 nm and 155-499 nm averaged for 10°-sectors. Distant shipping lanes sector, aging sector, nearby shipping lane sector, and harbor of Utö are marked with vertical line pairs.**





**Figure 9: A)** Average number size distribution of the plumes ($NSD_{pl}$) from the nearby shipping lanes sector (280-300°) and **B)** from the distant shipping lanes sector (70-130°) during the three FSC restriction periods. **C)** Change of $NSD_{pl}$ in nearby shipping lanes sector (280-300°) and **D)** in the distant shipping lanes sector (70-130°) caused by the changes of the FSC restriction.






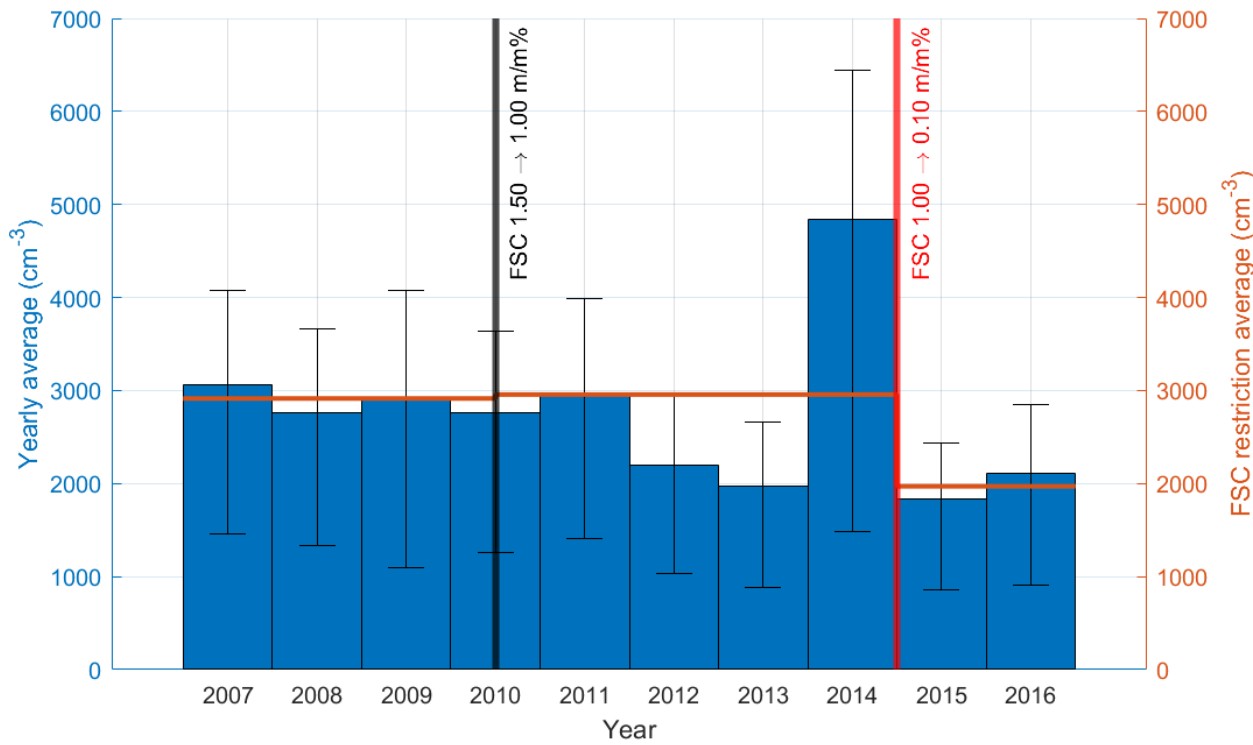

**Figure 10: Annual average total particle number concentrations (PNC_tot, blue bars) and the average PNC_tots (orange lines) during the different FSC restrictions. Changes of the FSC restrictions are marked with the black and red vertical lines. 25th and 75th percentiles are marked with the black error bars.**



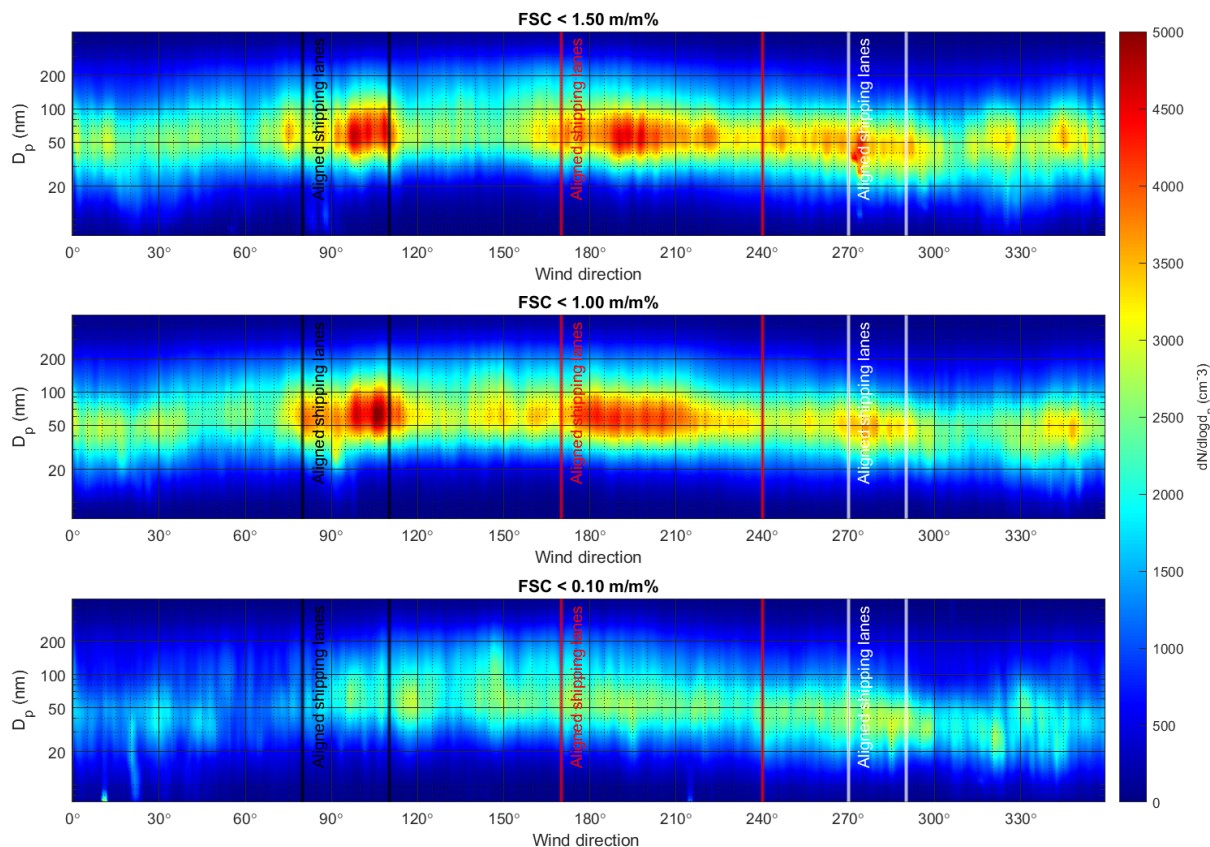


**Figure 11: Total number size distributions (NSD$_{tot}$) averaged for wind directions. Wind directions are shown on the x-axis, particle mobility diameters on the y-axis, and normalized particle number concentrations (dN/dlogd$_p$) with the colors. Directions, where there are shipping lanes at least partially aligned with the wind directions, are marked with vertical line pairs (black, red, and white). These shipping lanes have been estimated from Figure 1A.**


**Figure 12: Average number size distributions (NSD$_{pl}$) and volume size distributions (VSD$_{pl}$) of the plumes in the sector of 220-260° for three FSC restriction periods. A) nighttime NSD$_{pl}$, B) nighttime VSD$_{pl}$, C) daytime NSD$_{pl}$, D) daytime VSD$_{pl}$, E) change in NSD$_{pl}$ caused by exposure to sunlight, F) change in VSD$_{pl}$ caused by exposure to sunlight.**






**Table 1: Particle number (PNC$_{pl}$) and volume concentrations (VC$_{pl}$) of plumes during the three different sulfur restriction periods and the change in concentrations between day- and nighttimes.**

| Variable | PNC$_{pl}$ ($cm^{-3}$) | | | VC$_{pl}$ ($\mu m^3\ m^{-3}$) | | |
|---|---|---|---|---|---|---|
| FSC limit (m/m%) | 1.50 | 1.00 | 0.10 | 1.50 | 1.00 | 0.10 |
| Nighttime | 1910 | 1470 | 1200 | 0.527 | 0.422 | 0.373 |
| Daytime | 1400 | 1300 | 1090 | 0.585 | 0.458 | 0.402 |
| Change | -510 | -170 | -110 | 0.058 | 0.036 | 0.029 |
| Change (%) | -27 | -11 | -9 | +11 | +9 | +8 |