# Peer review of "Effects of marine fuel sulfur restrictions on particle number concentrations and size distributions in ship plumes at the Baltic Sea"

_Atmospheric Chemistry and Physics, 2020_

## Referee Comment (RC1) · Adam Kristensson (Referee) · 23 Oct 2020

General comments

I recommend the paper to be published subject to only minor revisions.

I find the paper to be very important for the scientific aerosol measurement community, since it presents data from shipping emissions. It also is very important for policy making, since reductions in FSC is seen as a drastic decrease in air pollution levels. Modelling community will also value this paper highly due to the FSC content influence and ship emission pollution data, which can be used to validate the models. Finally, it

gives inisights to how particles from ship emissions are ageing in the atmosphere.

It is very well written language wise, and presents all contents in a structured manner.

Specific comments

Chapter 3.2

The reason why the Zanatta et al. reference shows slightly higher ship plume contributions than your paper or the paper by Ausmeel et al. is not only due to meteorological influences or variations in NSD size ranges. Important factors are likely also that Zanatta et al. might be closer to the ships, and/or that they observe several shipping lanes simultaneously (they don't exclude superimposed ship plumes), and/or that they have very few measurements, so it can be a statistical effect as well. Please write these explanations as well.

Chapter 3.3.

In Figure 6 you see a dramatically higher plume concentration for the 200 degree wind direction as compared to the 270 degree wind direction. This is very interesting and should be included in the discussions about the results, because logically it makes little sense. So, what could be the explanation? If the ships at 200 degrees are about the same size and type as the ships at 270 degrees, then the explanation for the difference must be meteorological; for example, plumes from 270 degrees are relatively more diluted vertically, and/or they are transported higher up in the marine boundary layer, partly missing to descend to Utö station measurement height. But, if it is an effect of difference in ship types or sizes, you should roughly explain the difference in fleet composition between different 10 degree wind sectors, because this could explain why you see such different concentrations in different wind sectors.

Mass concentrations

You should estimate the average contribution of the ship plumes to PM0.15 and/or PM0.5 mass concentrations during the three different FSC regimes by assuming a

constant density of particles. This assessment of the importance of ship emissions to PM mass concentrations is extremely valuable for the scientific community, policy making, and health effects. These contributions can be compared to Ausmeel et al., 2020; Atmos. Chem. Phys., 20, 9135–9151, 2020. You have already calculated the volume concentrations, so should be straightforward to calculate the mass concentration by just multiplying with the density. Your contribution to PM0.5 is about a factor 7-8 higher than at Falsterbo peninsula, which is likely due to the fact that ships in Falsterbo are smaller, and possibly due to the long average distance to the shipping lane in Falsterbo of around 10 km – however, I don't know the average distance to ships in your paper.

Technical corrections

Introduction

Lin et al., 2018 is an epidemiological study, while Partanen et al., 2013 present only a calculated expected outcome for health effects with reduced ship emissions. Hence, you can claim that Lin et al. show that you have a link between ship emissions and health effects, while you can not write that Partanen et al. have linked reduced ship emissions to reduced disease or mortality burden. Please rephrase. The same reasoning goes for the phrasing: "The reduction of PM2.5 emissions from shipping has been shown to reduce the negative health effects". This hasn't been shown in the cited references, this has been estimated. Please rephrase. Otherwise, the reader might think that there is epidemiological evidence in all the referenced studies.

Figure 1.

It is hard to understand what is meant with "every 10000 shown" and "every 500th shown" and "every 10th shown". Please explain in a different way.

Chapter 3.2

Lines 186-187. It should be 700 and 1470 particles per cubic centimeter respectively for the Ausmeel study.

Chapter 3.3

Lines 244-245. You probably mean particles below 155 nm are not much affected by FSC instead of 134 nm, since they are actually affected at 134 nm?

Chapter 3.6

Line 331. It should read secondary aerosol (SA), and not secondary organic aerosol (SOA).

Conclusion

Line 396. You probably want to write >= 155 nm, and not > 155 nm?
* * *

---

## Referee Comment (RC2) · Anonymous Referee #2 · 17 Nov 2020

This manuscript presents a major data set of ship plumes, as observed at a fixed location downwind shipping lanes in a Baltic Sea. The analysis is based on measured particle number size distribution over a 10-year period, during which the sulfur content of the fuel used by marine ships decreased considerably. The paper is scientifically sound and original enough to warrant its publication. The text is well organized and clearly written. I have a few minor issues to be addressed before recommending the paper to be accepted for publication.

The authors should justify the selection of the size ranges 7-134 nm and 155-499 nm in their analysis. Why the border between these size ranges is well over 100 nm,

including both Aitken mode and the lower tail of the accumulation mode? Furthermore, the given borders of the size ranges are incorrect, because the middle points of size bins do not determine the upper and lower limits of the size ranges. For example, the upper limit of the first size range and lower limit of the second size range should be the same (somewhere between 134 and 155 nm, probably close to geometric mean of these two diameters).

Too much emphasis is given to the role of coagulation in shaping the observed particle number size distributions. It is true that coagulation is able to decrease total particle number concentrations in aging shipping plumes, but the reported particle number concentrations are way too low to cause any significant growth of the particles between their emissions and observations (times scales for coagulation growth are simply too short in these cases). This is related to discussion on lines 232-233, 281 and lines 412-413 (on these last lines, the authors give an impression that particle chemical composition would affect coagulation, which does not sound correct).

Why to use the concept total particle number size distribution mentioned on lines 306-307 and Figure 11? I would understand the word "total" if the particle population would have been treated somehow (e.g heating to different temperatures) and then compared to the non-treated situation, but this seems not to be the case here. Would just talking about particle number size distribution be enough?

What are the authors referring to with dilution-related processes on line 233? Changes in gas-phase chemistry and/or gas-particle partitioning of condensable vapors? Dilution itself does not affect particle size by any means.

Line 401: arriving at

Lines 425-426: Writing like this, the authors give a somewhat incorrect impression on aerosol influences on clouds. Cloud formation tends to be dominated meteorological processes (cooling of air, often in updrafts driven by other met phenomena). While aerosols particles could affect this at high aerosol loading due to aerosol-radiation

interactions, their main role is to modify cloud microphysical properties via aerosol-cloud interactions.

---

## Author Comment (AC1) · 29 Dec 2020

We thank the referees for taking the time to read the article and their valuable comments on our paper. To facilitate the revision process, we have copied the referee comments. The referee comments have been marked with RC1: and RC2: at the beginning of each comment, corresponding to referees 1 and 2 respectively. Responses to the reviewer comments are given after each referee comment beginning with AR: (author response). We have responded to all the referee comments and made alterations to our paper. Additionally, during the revision process, the authors noticed that wind directions were marked falsely for some of the data. The minor changes based

on incorrect wind directions, as well as other minor errors noticed during the revision, are listed at the end of this document.

Adam Kristensson (Referee 1)

adam.kristensson@nuclear.lu.se

General comments

RC1: I recommend the paper to be published subject to only minor revisions. I find the paper to be very important for the scientific aerosol measurement community, since it presents data from shipping emissions. It also is very important for policy making, since reductions in FSC is seen as a drastic decrease in air pollution levels. Modelling community will also value this paper highly due to the FSC content influence and ship emission pollution data, which can be used to validate the models. Finally, it gives inisights to how particles from ship emissions are ageing in the atmosphere. It is very well written language wise, and presents all contents in a structured manner.

Specific comments

Chapter 3.2

RC1: The reason why the Zanatta et al. reference shows slightly higher ship plume contributions than your paper or the paper by Ausmeel et al. is not only due to meteorological influences or variations in NSD size ranges. Important factors are likely also that Zanatta et al. might be closer to the ships, and/or that they observe several shipping lanes simultaneously (they don't exclude superimposed ship plumes), and/or that they have very few measurements, so it can be a statistical effect as well. Please write these explanations as well.

AR: The vicinity of the source in Zannatta et al. (2020) has been added to the text by informing that the measurements were made straight over a shipping lane. We

agree with the Referee that the shorter distance to the measured ships in Zanatta et al., (2020) and the sample size can both cause the difference in the reported concentrations. The text is modified and the following text is added to the manuscript: "Concentrations observed by Zanatta et al. (2020) were higher likely because of the fresher plumes (plume age ~200-800 s). Also, the small sample size (only 9 continuous periods of single or multiple exhaust plumes each lasting a couple of minutes at maximum) might cause statistical uncertainty to the results."

Chapter 3.3.

RC1: In Figure 6 you see a dramatically higher plume concentration for the 200 degree wind direction as compared to the 270 degree wind direction. This is very interesting and should be included in the discussions about the results, because logically it makes little sense. So, what could be the explanation? If the ships at 200 degrees are about the same size and type as the ships at 270 degrees, then the explanation for the difference must be meteorological; for example, plumes from 270 degrees are relatively more diluted vertically, and/or they are transported higher up in the marine boundary layer, partly missing to descend to Utö station measurement height. But, if it is an effect of difference in ship types or sizes, you should roughly explain the difference in fleet composition between different 10 degree wind sectors, because this could explain why you see such different concentrations in different wind sectors.

AR: Authors agree that the difference makes logically little sense and a short discussion on the subject is added to Chapter 3.3: "From angles 190-210°, the PNCpls were significantly higher than from angles 260-280°. The reason for this was investigated by calculating the fractions of specific ship types observed at these angles. These fractions are presented in the supplemental material (Fig. S2). No significant difference was observed except the higher faction of Roll-on/roll-off passenger vessels in angles 190-210°. Therefore, the difference in the concentrations is not likely to be related to different ship fleets but it can be caused by the meteorology of the area. The lower concentrations of the plumes in angles 260-280° might be caused by higher vertical

dilution or the plumes might be transported higher up in the marine boundary layer. However, the effect of meteorology could not be validated." Fig. S2 is added to the author comments as Fig. 1.

Mass concentrations

RC1: You should estimate the average contribution of the ship plumes to PM0.15 and/or PM0.5 mass concentrations during the three different FSC regimes by assuming a constant density of particles. This assessment of the importance of ship emissions to PM mass concentrations is extremely valuable for the scientific community, policy making, and health effects. These contributions can be compared to Ausmeel et al., 2020; Atmos. Chem. Phys., 20, 9135–9151, 2020. You have already calculated the volume concentrations, so should be straightforward to calculate the mass concentration by just multiplying with the density. Your contribution to PM0.5 is about a factor 7-8 higher than at Falsterbo peninsula, which is likely due to the fact that ships in Falsterbo are smaller, and possibly due to the long average distance to the shipping lane in Falsterbo of around 10 km – however, I don't know the average distance to ships in your paper.

AR: As suggested by the Referee, upper and lower estimates for the direct contribution of ship exhaust plumes to PM0.144 and PM0.537 during different FSC restrictions were calculated and compared to Ausmeel et al., (2020). A table including these values for valid plumes, all plumes, and ambient aerosol is added in the article (Table 1) with the following text at the end of section 3.5: "From the NSDpls and NSDtots, the masses of PM0.144 and PM0.537 (PM of particles smaller than 144 and 537 nm) were calculated. Both PM0.144 and PM0.537 were calculated for the valid plumes and all measured plumes. Also, the contributions of both to ambient PM0.144 and PM0.537 were calculated by taking into account the duration and number of the plumes (Table 1). Both only the valid and all plumes were calculated to give approximate lower and upper limits for the contributions as valid plumes almost certainly did not include all the plumes and all the measured plumes likely included also other phenomena in addition to plumes. A steady decrease of the contributions of the ship plumes to PM0.144

and PM0.537 can be seen after the FSC restriction changes with the contributions reducing overall from 5.5 - 14.0 % (31.6 - 80.8 ng m-3) to 3.9 - 8.9 % (12.9 – 29.8 ng m3) for PM0.144 and from 2.8 - 7.4 % (106.3 – 283.3 ng m-3) to 2.4 - 5.5 % (60.0 - 136.2 ng m-3) for PM0.537. Similar values of 34 $\pm$ 19 ng m-3 (summer) and 18 $\pm$ 8 ng m-3 (winter) for PM0.15 during the FSC restriction of 0.10 m/m% have been reported by Ausmeel et al., (2020). Contribution to PM0.5 reported by Ausmeel et al., (2020) is slightly lower compared to this study, 37 $\pm$ 20 ng m-3 (summer) and 29 $\pm$ 13 ng m-3 (winter). An important thing to note is that these contributions are only the direct contributions of the detected plumes to ambient PM0.144 and PM0.537. The real contributions are likely to be higher because of the contribution from ship plumes diluted beyond detection. In calculating the masses of plume particles, a density of 1.23 g cm-3 was used, corresponding to the density of ship exhaust particles calculated based on the particle chemistry (Petzold et al., 2008). For ambient aerosol, a density of 1.10 g cm-3 was used. This is the average of the effective density values reported by Geller et al. (2006) for coastal aerosols of different sizes. They reported density values of 1.19, 1.14, 0.99, and 1.06 g cm-3 for particle sizes of 50, 118, 146, and 202 nm, respectively." Table 1 is added to the author comments as Fig. 2. The following reference is added to the reference list: Geller, M., Biswas, S., and Sioutas, C.: Determination of Particle Effective Density in Urban Environments with a Differential Mobility Analyzer and Aerosol Particle Mass Analyzer, Aerosol Sci. Technol., 40:9, 709-723, https://doi.org/10.1080/02786820600803925, 2006.

Technical corrections

Introduction

RC1: Lin et al., 2018 is an epidemiological study, while Partanen et al., 2013 present only a calculated expected outcome for health effects with reduced ship emissions. Hence, you can claim that Lin et al. show that you have a link between ship emissions and health effects, while you can not write that Partanen et al. have linked reduced ship emissions to reduced disease or mortality burden. Please rephrase. The same

reasoning goes for the phrasing: "The reduction of PM2.5 emissions from shipping has been shown to reduce the negative health effects". This hasn't been shown in the cited references, this has been estimated. Please rephrase. Otherwise, the reader might think that there is epidemiological evidence in all the referenced studies.

AR: The references are corrected as suggested. Partanen et al., 2013 is removed from the reference in the sentence: "PM from shipping emissions can also be transported hundreds of kilometers inland (Lv et al., 2018), and have been linked to increased cardiovascular mortality and morbidity (Lin et al., 2018, Partanen et al., 2013)", and the sentence: "The reduction of PM2.5 emissions from shipping has been shown to reduce the negative health effects (Barregard et al., 2019; C. Chen et al., 2019; Partanen et al., 2013; Sofiev et al., 2018)." is changed to: "The reduction of PM2.5 emissions from shipping has been estimated to reduce the negative health effects (Barregard et al., 2019; Chen et al., 2019; Partanen et al., 2013; Sofiev et al., 2018)."

RC1: It is hard to understand what is meant with "every 10000 shown" and "every 500th shown" and "every 10th shown". Please explain in a different way.

AR: The indistinct expressions are removed from the caption and replaced with the following text: "(0.01, 0.2, and 10 % of the locations plotted for figures A, B, and C respectively)"

Chapter 3.2

RC1: Lines 186-187. It should be 700 and 1470 particles per cubic centimeter respectively for the Ausmeel study.

AR: The text has been modified by replacing: 750 with 700. Corrected sentence: "PNCpl attained in this work are similar to the concentrations of plume observed in similar studies; The median PNC of 700 and 1470 cm-3 were measured during winter and summertime, respectively, in southern Sweden (Ausmeel et al. 2019)."

Chapter 3.3

RC1: Lines 244-245. You probably mean particles below 155 nm are not much affected by FSC instead of 134 nm, since they are actually affected at 134 nm?

AR: Authors agree that the notation is misleading as the concentrations at 134 nm particle size are affected. As Referee 2 pointed out, the limits for the size bins have been reported incorrectly with bin midpoints instead of limits. Therefore, the correct limit of 144 nm is changed into the text: "Therefore, the effect of sulfur restrictions seems to be mostly limited to the PNCpl in the particle sizes smaller than 144 nm."

Chapter 3.6

RC1: Line 331. It should read secondary aerosol (SA), and not secondary organic aerosol (SOA).

AR: Secondary organic aerosol SOA has been changed to secondary aerosol SA in the text.

Conclusion

RC1: Line 396. You probably want to write >= 155 nm, and not > 155 nm?

AR: As Referee 2 suggested, the notation of bin limits is changed from bin midpoint (155 nm) to the lower limit of the size bin (144 nm). Therefore, the notation is corrected now to: "$\geq$ 144 nm"

Anonymous Referee 2

RC2: This manuscript presents a major data set of ship plumes, as observed at a fixed location downwind shipping lanes in a Baltic Sea. The analysis is based on measured particle number size distribution over a 10-year period, during which the sulfur content of the fuel used by marine ships decreased considerably. The paper is scientifically sound and original enough to warrant its publication. The text is well organized and clearly written. I have a few minor issues to be addressed before recommending the

paper to be accepted for publication.

RC2: The authors should justify the selection of the size ranges 7-134 nm and 155-499 nm in their analysis. Why the border between these size ranges is well over 100 nm, including both Aitken mode and the lower tail of the accumulation mode? Furthermore, the given borders of the size ranges are incorrect, because the middle points of size bins do not determine the upper and lower limits of the size ranges. For example, the upper limit of the first size range and lower limit of the second size range should be the same (somewhere between 134 and 155 nm, probably close to geometric mean of these two diameters).

AR: As suggested by the Referee, we changed bin midpoints to the bin limits in the text. The limits of 7-134 nm and 155-499 nm (after corrections 7-144 nm and 144-537 nm) were chosen as they were found to correspond well to the seen effects of the FSC restrictions on the number size distribution of the plumes. To illustrate this the (Fig. S3) was added in the supplemental data. This figure is added to the authors comments as Fig. 3. Also, the following text is added to Chapter 3.3: "The size ranges were chosen based on testing to find a limit for the effects of the FSC restrictions on PNCpls. Average NSDpls for all the plumes during different sulfur restrictions are presented in the supplemental material (Fig. S3). From these distributions, it is visible that the particle size of 144 nm acts as a limit for the effects of the FSC restrictions."

RC2: Too much emphasis is given to the role of coagulation in shaping the observed particle number size distributions. It is true that coagulation is able to decrease total particle number concentrations in aging shipping plumes, but the reported particle number concentrations are way too low to cause any significant growth of the particles between their emissions and observations (times scales for coagulation growth are simply too short in these cases). This is related to discussion on lines 232-233, 281 and lines 412-413 (on these last lines, the authors give an impression that particle chemical composition would affect coagulation, which does not sound correct).

AR: Authors agree that the coagulation has been given too much emphasis as it is likely to affect distributions only at high concentrations and therefore plays only a minor role in the processing of diluted ship exhaust plumes in the atmosphere. The discussion on coagulation is significantly reduced in the text mostly by deleting references to coagulation altogether. In Chapter 3.6 the sentence: "Coagulation cannot explain the higher number of smaller particles observed in Fig. 12, but the evaporation of liquid matter from the surface of medium-sized particles exposing the small solid core particles could explain this." Is changed to: "The higher number of smaller particles observed in Fig. 12, can be explained by the evaporation of liquid matter from the surface of medium-sized particles exposing the small solid core particles."

RC2: Why to use the concept total particle number size distribution mentioned on lines 306- 307 and Figure 11? I would understand the word "total" if the particle population would have been treated somehow (e.g heating to different temperatures) and then compared to the non-treated situation, but this seems not to be the case here. Would just talking about particle number size distribution be enough?

AR: Authors agree that the usage of the term "total" with the ambient particle number concentrations is unnecessary. Therefore "total" terms related to ambient particle number concentrations were either removed or changed to "ambient" in the text. Subscript tot was left unchanged and it refers to ambient particle number concentrations.

RC2: What are the authors referring to with dilution-related processes on line 233? Changes in gas-phase chemistry and/or gas-particle partitioning of condensable vapors? Dilution itself does not affect particle size by any means.

AR: Whole sentence has been removed as unnecessary.

RC2: Line 401: arriving at

AR: Corrected as suggested.

RC2: Lines 425-426: Writing like this, the authors give a somewhat incorrect impression on aerosol influences on clouds. Cloud formation tends to be dominated meteorological processes (cooling of air, often in updrafts driven by other met phenomena). While aerosols particles could affect this at high aerosol loading due to aerosol-radiation interactions, their main role is to modify cloud microphysical properties via aerosol-cloud interactions.

AR: Authors agree that the impression given by the text is somewhat incorrect and therefore we replaced the lines 423-427 with the following text: "The reduction of particle concentration in the size range of $\sim$ 33-144 nm observed in this study implies significant effects on radiative properties of marine clouds. Particle diameter range from 40 to 120 nm has been identified to be crucial for cloud condensation nuclei, as larger particles are almost always activated to form cloud droplets, while particles smaller than 40 nm are not activated with realistic supersaturations (Dusek et al., 2006). Therefore, ships running with low-sulfur fuel produce fewer cloud condensation nuclei in this critical size range. This can lead to a substantial inverse Twomey effect, increasing the albedo of marine clouds, and thus, having an indirect warming effect on the climate (Twomey, 1977)." The following reference is added to the reference list: Twomey, S.: The influence of Pollution on the Shortwave Albedo of Clouds, J. Atmos. Sci., 34, 1149-1152, https://doi.org/10.1175/1520-0469(1977)034<1149:TIOPOT>2.0.CO;2, 1977.

Other changes

Falsely marked winds

When calculating new limits for the particle size ranges, the authors noticed that wind directions were marked falsely for some of the data. In data extracted from the database wind direction 0 was interpreted by authors as the wind from the north, while in reality that meant still air (wind speed 0, thus no wind direction). Also, in some figures, presenting different properties of plumes and ambient aerosol from different wind directions both upper and lower limits of each of the size bins were included in the sectors. This resulted in some of the data being counted twice. These minor mistakes

were fixed, and the corrections made to the text are explained in more detail later. Overall, these corrections changed the results only minimally and had no impact on conclusions as still wind counted only for 0.34% of the measurement time. Below the detailed list of corrections made to figures:

-Figure 1: The wind rose in 1C included still winds (wind direction 0) counted as north wind. This was corrected.

-Figure 4: Both diagrams included still winds counted as north wind this was corrected.

-Figure 6: During all the three FSC restrictions the still winds were counted as north winds. This was corrected. Also, in each 10-degree sector, both the upper and lower sector limits were included in each sector. This resulted in the plumes exactly from angle 10, 20, 30, 40.. and so on, being counted twice. This was corrected by only including the lower limit in each sector.

-Figure 7: During all the three FSC restrictions the still winds were counted as north winds. This was corrected. Also, in each 10-degree sector, both the upper and lower sector limits were included in each sector. This resulted in the plumes exactly from angle 10, 20, 30, 40.. and so on, being counted twice. This was corrected by only including the lower limit in each sector.

-Figure 8: During all the three FSC restrictions the still winds were counted as north winds. This was corrected. Also, in each 10-degree sector, both the upper and lower sector limits were included in each sector. This resulted in the plumes exactly from angle 10, 20, 30, 40.. and so on, being counted twice. This was corrected by only including the lower limit in each sector. The new size bin limits instead of midpoints were changed to the titles.

-Figure 9: In the data from which the plumes were separated the still winds were counted as north winds. This could have affected some of the plumes in degrees 280-300° and 70-130° through averaging the wind directions for the plumes. This was

corrected.

-Figure 11: In addition to correcting wind angles, it was noticed that Fig. 11 was previously calculated using only data from valid time periods. This was changed as the validity is meaningful only in analyzing the plumes. Therefore, now in Fig. 11, all the measured data is used. These corrections resulted in minor changes in the directions of the highest concentrations. In Chapter 3.5 the angles with elevated particle concentrations are adjusted according to the new Fig 11. (80-120° to 90-120°, 160-230° to 170 -220° and 260-290° to 270-290°). Also, the angles 270-290° now show slightly less elevated concentrations and this is mentioned in the text referring to angles 270-290° only as elevated, not highest, concentrations.

-Figure 12: In the data from which the plumes were separated the still winds were counted as north winds. This could have affected some of the plumes in degrees 220-260° through averaging the wind directions for the plumes. This was corrected. Also, the titles of Figures 12 C and 12 D were corrected to now show correctly > 200, not < 0.

-Figure S2: (now S4). During all the three FSC restrictions the still winds were counted as north winds. This was corrected. Also, in each 10-degree sector, both the upper and lower sector limits were included in each sector. This resulted in the plumes exactly from angle 10, 20, 30, 40.. and so on, being counted twice. This was corrected by only including the lower limit in each sector. The new size bin limits instead of midpoints were changed to the titles.

-Figure S4 (now S6): Still winds were counted as north winds. This was corrected. Also, in each 10-degree sector, both the upper and lower sector limits were included in each sector. This resulted in the plumes exactly from angle 10, 20, 30, 40.. and so on, being counted twice. This was corrected by only including the lower limit in each sector. Also, the caption was modified to include the limit for the counts: "The black line marks the limit of 40 observed plumes per 10° sectors." Also, the criteria was changed from

50 to 40 plumes per 10-degree sector because of the changes in the plume counts caused by wind data corrections.

-Figure S5 (now S7): During all the three FSC restrictions the still winds were counted as north winds. This was corrected. Also, in each 10-degree sector, both the upper and lower sector limits were included in each sector. This resulted in the plumes exactly from angle 10, 20, 30, 40.. and so on, being counted twice. This was corrected by only including the lower limit in each sector.

Other corrected mistakes and changes made that are not listed earlier

-Minor typing mistakes corrected (shown with track changes in manuscript).

-Table 1 number changed to 2.

-Model of the CPC was added to chapter 2.3: "The model of the CPC was TSI model 3010."

- A wrong email (aman@email.com) for the corresponding author Sami Seppälä was given in the supplemental data and it was corrected to sami.seppala@fmi.fi

-In Chapter 2.3 the sentence: "During the measurements, the DMA and CPC were changed a few times due to malfunctions." Was changed to: "During the measurements, parts of the DMPS including the CPC were changed a few times due to malfunctions."

-The following sentence was added to chapter 2.3: "DMA losses were assumed to be 15% independent of the particle size based on the laboratory tests."

-All CMD:s (count median diameters) in the text are changed to GMDs (geometric mean diameters) as the used equation is in reality equation for GMD.

- Additional funders added Acknowledgements and it now reads: "Acknowledgements. This work was financed by the European Research Infrastructure for the observation of Aerosol, Clouds, and Trace Gases (ACTRIS), the European Union's Horizon

2020 Programme Research and Innovation action under grant agreement no. 814893, SCIPPER, Academy of Finland Center of Excellence programme (grant no. 307331), Academy of Finland Flagship funding (grant no. 337552) and Academy of Finland Flagship funding Atmosphere and Climate Competence Center, ACCC (grant no. 337549)."
* * *
[Figure]

**Fig. 1.** Figure S2: Fractions (%) of different ship types from angles 190-210° and 260-280°. LNG and RoRo stand for liquefied petroleum gas and Roll on roll of, respectively.

| Variable | $PM_{0.144}$ | | | $PM_{0.537}$ | | |
|---|---|---|---|---|---|---|
| FSC limit (m/m%) | 1.50 | 1.00 | 0.10 | 1.50 | 1.00 | 0.10 |
| Average MC of valid plumes (µg cm$^{-3}$) | 0.20 | 0.18 | 0.11 | 0.68 | 0.64 | 0.51 |
| Contribution of valid plumes to ambient MC (%) | 5.5 | 4.9 | 3.9 | 2.8 | 2.6 | 2.4 |
| Average MC of all plumes (µg cm$^{-3}$) | 0.22 | 0.21 | 0.12 | 0.78 | 0.76 | 0.55 |
| Contribution of all plumes to ambient MC (%) | 14.0 | 12.1 | 8.9 | 7.4 | 6.7 | 5.5 |
| Average ambient MC (µg cm$^{-3}$) | 0.58 | 0.58 | 0.33 | 3.81 | 3.68 | 2.47 |

**Fig. 2.** Table 1: Average mass concentrations (MC) of ambient aerosol, valid plumes, and all plumes during the different fuel sulfur content (FSC) restrictions and the contributions of valid and all the plumes

**Fig. 3.** Figure S3: Average number size distributions of the valid plumes (NSDpl) during the different fuel sulfur content (FSC) restrictions.